# Safety and Efficacy of Intravenous and Intrathecal Delivery of AAV9-Mediated ARSA in Minipigs

**DOI:** 10.3390/ijms24119204

**Published:** 2023-05-24

**Authors:** Aysilu Mullagulova, Alisa Shaimardanova, Valeriya Solovyeva, Yana Mukhamedshina, Daria Chulpanova, Alexander Kostennikov, Shaza Issa, Albert Rizvanov

**Affiliations:** 1Institute for Fundamental Medicine and Biology, Kazan Federal University, 420008 Kazan, Russia; ayimullagulova@kpfu.ru (A.M.); alisashajmardanova@kpfu.ru (A.S.); vavsoloveva@kpfu.ru (V.S.); yomuhamedshina@kpfu.ru (Y.M.); daschulpanova@kpfu.ru (D.C.); aleakostennikov@kpfu.ru (A.K.); shaza.issa98@outlook.com (S.I.); 2Department of Histology, Cytology, and Embryology, Kazan State Medical University, 420012 Kazan, Russia; 3Department of Genetics and Biotechnology, St. Petersburg State University, 199034 St. Petersburg, Russia

**Keywords:** adeno-associated virus, metachromatic leukodystrophy, arylsulfatase A, gene therapy, central nervous system, peripheral nervous system, neurodegeneration

## Abstract

Metachromatic leukodystrophy (MLD) is a hereditary neurodegenerative disease characterized by demyelination and motor and cognitive impairments due to deficiencies of the lysosomal enzyme arylsulfatase A (ARSA) or the saposin B activator protein (SapB). Current treatments are limited; however, gene therapy using adeno-associated virus (AAV) vectors for ARSA delivery has shown promising results. The main challenges for MLD gene therapy include optimizing the AAV dosage, selecting the most effective serotype, and determining the best route of administration for ARSA delivery into the central nervous system. This study aims to evaluate the safety and efficacy of AAV serotype 9 encoding ARSA (AAV9-ARSA) gene therapy when administered intravenously or intrathecally in minipigs, a large animal model with anatomical and physiological similarities to humans. By comparing these two administration methods, this study contributes to the understanding of how to improve the effectiveness of MLD gene therapy and offers valuable insights for future clinical applications.

## 1. Introduction

Metachromatic leukodystrophy (MLD) is an autosomal recessive hereditary neurodegenerative disease that belongs to the group of lysosomal storage diseases (LSDs). It is characterized by damage to the myelin sheath, which covers most of the nerve fibers in the central (CNS) and peripheral nervous systems (PNS). MLD is caused by deficiencies in the lysosomal enzyme arylsulfatase A (ARSA) (OMIM: 250100) or the saposin B activator protein (SapB) (OMIM: 249900). Clinically, the disease manifests as progressive motor and cognitive impairments [1]. MLD is one of the most common leukodystrophies, with an incidence rate of 1:40,000. However, in some isolated populations, the incidence rate can be much higher, such as 1:75 among Habbani Jews and 1:2500 among the Navajo [2].

The term MLD refers to the presence of metachromatic granules in affected cells, which form due to the accumulation of sulfatides and sphingolipids found in myelin. In MLD, sulfatides build up in oligodendrocytes, microglia, CNS neurons, Schwann cells, PNS macrophages, and cells in various internal organs [3,4,5,6,7,8,9]. Demyelination in MLD results in impaired motor function, spastic tetraparesis, ataxia, convulsions, optic nerve atrophy, and cognitive impairment [10,11]. The exact mechanisms of the demyelination remain unknown, but it is believed that increased levels of sulfatides and decreased levels of their cleavage products cause instability in the myelin sheath, ultimately leading to demyelination [12]. Furthermore, the accumulation of sulfatides on the endoplasmic reticulum (ER) membrane triggers calcium release into the cytoplasm, causing changes in calcium homeostasis that lead to cellular stress and apoptosis [8]. The accumulation of sulfatides leads to neuronal degeneration and astrocyte dysfunction and triggers an inflammatory response in MLD patients. Both the plasma and cerebrospinal fluid (CSF) show elevated levels of monocyte chemoattractant protein 1 (MCP-1), interleukin 1 receptor antagonist (IL-1Ra), IL-8, macrophage inflammatory protein 1β (MIP-1β), and vascular endothelial growth factor (VEGF) [13]. In MLD, there may not be a correlation between demyelination and the presence of metachromatic material. Consequently, complement activation via an alternative pathway exacerbates myelin damage in MLD by inducing or enhancing an anti-myelin immune response [14]. Sulfatide accumulation and demyelination in the PNS can prompt the release of inflammatory cytokines, activate endoneurial macrophages, and recruit inflammatory myeloid cells and lymphocytes from the periphery. These processes contribute to apoptosis and can lead to the progression of demyelination and neuroinflammation, which are also observed in some other metabolic and neurodegenerative diseases [15].

A residual enzymatic activity of 10–15% can be sufficient for sulfatide degradation and the maintenance of a normal life in ARSA deficiency [16]. There is a pseudo-deficiency of ARSA, where the enzyme has low enzymatic activity (about 5–20%) but no disease phenotype is observed, as the residual activity is sufficient to catabolize its natural substrate [17]. In some instances, a shortened protein can be synthesized without a change in activity [18]. Based on residual enzymatic activity levels, MLD presents in various forms that differ in their ages of clinical manifestation and rates of disease progression: infantile (late infantile, from 0 to 4 years) [19,20], juvenile (from 4 to 15 years old) [21,22,23], and the adult form (over 15 years old) [21,24,25,26]. The late infantile form is the most common, occurring in 50–60% of all patients, while 20–30% are diagnosed with the juvenile form, and the adult form is the rarest, affecting about 15–20% of MLD patients [1].

Currently, there is no effective treatment for MLD. Clinical cases involving bone marrow transplantation (BMT), hematopoietic stem cells (HSC), or umbilical cord blood transplantation have been reported, but the therapeutic efficacy of these approaches remains insufficient to prevent the worsening of neurological symptoms in patients [27,28]. Promising results have been achieved using gene therapy methods to deliver the wild-type *ARSA* gene within vectors based on various adeno-associated virus serotypes (AAV) [29,30] as well as using mesenchymal stem cells [31,32] and combined gene–cell therapy [33]. A primary challenge in treating diseases affecting the nervous system is the poor permeability of the blood–brain barrier (BBB), which limits a drug’s access to the nervous system when administered systemically and consequently reduces the effectiveness of many therapeutic approaches [34,35]. Moreover, the successful distribution of therapeutic drugs throughout the nervous system, including the PNS, is crucial for preventing the progression of MLD.

AAVs are promising vectors for ARSA delivery into neurons. AAVs can trans-synaptically transduce neurons over a wide range from the injection site through anterograde transport [36]. However, some studies have reported that anterograde transport down the axon is limited to specific serotypes, such as AAV5 [37], but is not observed in AAV1, AAV2, AAV6, AAV8, or AAV9, while other studies have demonstrated the anterograde transport of AAV1, AAV2, and AAV6 [36]. AAV5-ARSA administration into the brains of MLD mice resulted in prolonged expression of the *ARSA* gene (3–15 months), facilitating an almost complete disappearance of pathological changes in the brain and preventing sulfatide accumulation [38]. Promising results have also been obtained using AAV9-encoding *ARSA* and the green fluorescent protein reporter gene. Injecting the viral vector into the jugular veins of newborn MLD mice led to prolonged expression of the *ARSA* gene (up to 15 weeks), primarily in muscle and heart cells, with moderate expression also found in CNS cells. In treated mice, sulfatide accumulation was significantly reduced in the brain and spinal cord, and their levels showed no differences compared to wild-type mice [30]. Among new AAV serotypes, AAVrh.10, isolated from non-human primates, migrated more efficiently from the intracerebral site compared to AAV1, AAV2, AAV5, AAV7, or AAV8 [39,40]. Injections of AAVrh.10-ARSA into the brains of 8-month-old MLD mice corrected the accumulation of certain sulfatides in oligodendrocytes, as this AAV serotype can apparently infect oligodendrocytes (up to 9%) [40]. In a phase I/II clinical trial investigating this virus (NCT01801709), four children in the pre-symptomatic or very early symptomatic stages received 12 injections of 1 × 10^12^ or 4 × 10^12^ (depending on age) AAVrh.10-ARSA-transducing units into the brain’s white matter. The ARSA activity in the CSF, which had been undetectable before treatment, was reported to reach 20–70% of control values after the injections. However, the symptoms in the early-stage patients continued to worsen, and the asymptomatic patients developed MLD that was not significantly different from the natural course of the disease [29]. AAV-PHP.eB has recently been described as a highly efficient serotype for crossing the BBB following intravenous delivery, providing efficient transduction of the brain and spinal cord in GFAP-Cre mouse models expressing Cre under the control of the mouse GFAP promoter and in C57Bl/6J mice [41,42]. A single intravenous administration of AAVPHP.eB-hARSA-HA into an MLD mouse model resulted in the stable expression of the ARSA enzyme in the brains and spinal cords of experimental animals. Audouard et al. demonstrated a complete correction of sulfatide accumulation in the spinal cords of model animals [43], which was not observed when AAVrh10-ARSA was injected into the brain [40].

MLD is characterized by various clinical manifestations, and effective disease modeling is critical for studying its genetic basis, progression, diagnosis, and therapy [1]. However, there are no large animal models currently available to study MLD. Each model has its own advantages and limitations. Pigs possess several distinct advantages that make them suitable for research as an animal model. They are large, enabling easy puncture procedures, and have highly developed CNS and PNS, in addition to sharing important anatomical and physiological similarities with the human body.

Consequently, MLD gene therapy faces several challenges; addressing these would significantly increase its effectiveness. Further optimization of the administered AAV dose is crucial, along with the selection of the most effective serotype and route of administration for ARSA delivery into the CNS. In this study, a comparative analysis was conducted to assess the safety and efficacy of gene therapy using AAV9-ARSA when administered intravenously or intrathecally into minipigs.

## 2. Results

### 2.1. Codon Optimization

Codon optimization enhances the efficiency of the translation of mRNA into polypeptides, thus improving the effectiveness of expression vectors, including those used in gene therapy. Ribosomes translate codons with higher frequency faster, as they are recognized by tRNA molecules that are abundant in the cell. Therefore, the optimal codons used for the recipient organism’s genes are the most frequently occurring synonymous codons. This modification is believed to increase the expression efficiency of the therapeutic gene.

To create the plasmid vector, the codon composition of the cDNA nucleotide sequence of the *ARSA* gene was optimized using the OptimumGene™ algorithm (GenScript, Piscataway, NJ, USA). The mRNA nucleotide sequence of the human ARSA gene (NM_000487.6) was used as a template for codon optimization. Optimizing the codon composition allowed for the highest possible level of gene expression. The wild-type *ARSA* gene contains tandem rare codons that can reduce translation efficiency or even disable the translation mechanism. Codon optimization increased the codon adaptation index (CAI) of the *ARSA* gene from 0.83 to 0.90, with a CAI of 1.0 considered desirable for the highest gene expression level. Additionally, the content of GC pairs was increased from 61.33 to 65.45 to enhance the stability of the *ARSA* gene mRNA. The optimization process also changed negative cis-acting sites, but the amino acid sequence of the *ARSA* gene remained the same, at 481 amino acid residues.

The codon-optimized cDNA of the *ARSA* gene was synthesized and cloned into the pAAV-MCS plasmid vector (Agilent Technologies, Santa Clara, CA, USA) by GenScript (USA) (Appendix A). The resulting plasmid construct was then transformed into E. coli (TOP10), and several clones were selected for the isolation of plasmid DNA. To confirm the successful cloning, the isolated plasmid DNA was subjected to a restriction analysis using the BamHI restriction enzyme. The resulting plasmid, named pAAV-ARSA, was found to have the expected size of 6226 bp (Appendix A).

To confirm the functionality of the pAAV-ARSA genetic construct, an immortalized line of primary human embryonic kidney cells (HEK293T) was transfected using the TurboFect transfection agent (Thermo Fisher Scientific Inc., Waltham, MA, USA) according to the manufacturer’s recommended protocol. The transfection efficiency was assessed by measuring the enzyme activity and via a Western blot analysis. After 48 h of transfection, HEK293T-pAAV-ARSA showed a 15-fold increase in ARSA enzymatic activity compared to native cells (Figure 1a). Additionally, the protein was detected at a molecular weight of approximately 33 kDa (Figure 1b).

To confirm the functionality of the AAV9-ARSA genetic construct, HEK293T cells were transfected with protamine sulfate. The efficiency of the in vitro recombinant protein expression was confirmed via activity testing and a Western blot analysis. The enzymatic activity in the genetically modified HEK293T cells was found to be 2-fold higher compared to the native and modified AAV9-GFP cells, indicating the successful transduction and overexpression of the ARSA enzyme in the modified HEK293T cells (Figure 2a). The Western blot analysis confirmed the presence of the ARSA protein in the modified HEK293T cells, with a molecular weight of approximately 33 kDa (Figure 2b). Furthermore, the purity analysis of the vector using SDS-PAGE electrophoresis indicated that the AAV9-ARSA construct was pure, without any impurities. The viral proteins were shown to be viral protein 1 (VP1), with a molecular weight of 81.4 kDa; viral protein 2 (VP2), with a molecular weight of 66.3 kDa; and viral protein 3 (VP3), with a molecular weight of 59.8 kDa, with VP3 predominating, as expected (Figure 2c).

### 2.2. Analysis of ARSA Enzymatic Activity

The animals were divided into three groups. The first group received an intrathecal administration of AAV9-ARSA at a dose of 1 × 10^12^ genomic copies/kg, the second group received an intravenous administration at a dose of 3.77 × 10^13^ genomic copies/kg, and the third group served as a control with no AAV9-ARSA administration. The functionality of the recombinant AAV9-ARSA was tested by measuring the ARSA activity in the plasma, CSF, and various CNS structures.

No significant differences in the ARSA enzymatic activity were observed in the plasma of animals from different groups. However, following the intrathecal administration of AAV9-ARSA, the ARSA enzymatic activity in porcine CSF increased. Specifically, on days 7, 14, 21, and 35, significant increases in the ARSA enzymatic activity in the CSF of 146%, 169%, 153%, and 138%, respectively, were observed (Figure 3).

Increases of 154%, 255%, and 357% in the ARSA enzymatic activity were found, compared to the control group, in the cerebellum and the cervical and lumbar spinal cord, respectively, following the intravenous administration of AAV9-ARSA (Figure 4). These results indicate that neuronal cells began to express the functional enzyme in vivo following genetic modification by AAV9-ARSA, and the increase was observed in both groups of animals.

In the first group of experimental animals, after the intrathecal injection of AAV9-ARSA, mRNA copies of the *ARSA* gene were detected in different regions of the CNS. Specifically, 91,774, 11,181, 3883, and 15,144 copies of *ARSA* gene mRNA per μg of total RNA were detected in the cerebellum and the cervical, thoracic, and lumbar spinal cord, respectively (Figure 5a).

In the second group of experimental animals, after the intravenous administration of AAV9-ARSA, codon-optimized *ARSA* gene overexpression was only observed in the thoracic spinal cord and the dorsal root ganglia of the thoracic region. Specifically, 1104 copies of *ARSA* gene mRNA were detected per µg of total RNA in the thoracic spinal cord, while in the ganglia of the dorsal roots of the cervical, thoracic, and lumbar regions, 203, 666, and 185 copies of *ARSA* gene mRNA per μg of total RNA, respectively, were detected (Figure 5b).

### 2.3. Biochemical Blood Analysis

The biochemical parameters of minipigs’ blood sera were assessed after AAV9-ARSA administration. The ALT and creatinine-J levels did not show any changes after AAV9-ARSA administration. A statistically significant decrease in AST levels was observed on day 35 in the second group of animals, which received an intravenous administration. However, in the first group of animals, which received intrathecal injections, the AST levels remained unchanged (Figure 6). In the second group of animals, the bilirubin levels were significantly lower on days 7 and 35 following the intravenous administration of the drug, while no changes in the total bilirubin levels were observed in the first group of animals, which received an intrathecal administration of the drug.

The inflammatory cytokine profiles of the porcine serum and CSF were also studied. No statistically significant differences were detected between the levels of inflammatory cytokines and chemokines in different animal groups in the minipigs’ blood serum. In the CSF, increases in the IL-1ra levels were detected following both intrathecal (0.288 ± 0.257 pg/mL) and intravenous (0.078 ± 0.057 pg/mL) AAV9-ARSA administration, while no IL-1ra was detected in the CSF of the animals in the control group (Figure 7).

### 2.4. Pathomorphological Analysis

A comparative pathomorphological analysis of the kidney, heart, spleen, lung, and liver tissues was performed for both the control and experimental groups. The results did not reveal any significant pathomorphological changes in the studied organs as a result of the intrathecal or intravenous administration of AAV9-ARSA (Figure 8). A microscopic examination of the kidneys and hearts in the study’s groups did not reveal any foci of hemorrhage or other circulatory disorders, the deformation of renal glomeruli or muscle fibers, stromal edema, or leukocyte infiltration. A pathological analysis of the spleen did not reveal any signs of a hyperimmune response, such as hyperplasia or plethora, or signs of immunosuppression, such as the atrophy of lymphoid follicles, the desolation of the spleen pulp, or splenic anemia. A microscopic examination of the lung tissue showed that the parenchyma of the lung alveoli was lined with flattened alveolar epithelium, thin partitions were located between them without signs of edema, and no erythrostasis or diapedesis hemorrhages were found in the capillaries. Unfortunately, microscopic sections of the liver could not be obtained due to technical issues.

### 2.5. Assessment of ARSA Expression in Nervous Tissue

As a result of the immunofluorescence analysis, Purkinje neurons overexpressing ARSA were found in the cerebellar cortex in both experimental groups of animals but not in the control group (Figure 9A–A2). The number of these cells was higher in the group with intrathecal drug administration (45 ± 40.8) compared to the intravenous administration group (10.5 ± 4.3). It should be noted that in the second experimental group, Purkinje neurons overexpressing ARSA were mostly localized singly within one gyrus (Figure 9A2), while in control group, ARSA expression was lower and was mainly localized in the periphery of the cytoplasm (Figure 9A).

An analysis of the occipital cortex showed increased expression of ARSA in the subarachnoid spaces in minipigs from both experimental groups compared to the control group (Figure 9B–B2). However, an analysis of the average ARSA glow intensity in this area did not reveal significant differences between the experimental groups. In minipigs of both the experimental and control groups, ARSA+ cells were also found in the cerebral cortex. The specific luminescence in these cells was localized in the peripheral cytoplasm of the cell body and partially in the processes.

An analysis of transverse sections of the cervical, thoracic, and lumbar spinal cord showed no significant differences in ARSA expression in the white and gray matter in animals of the experimental and control groups. In the white matter, ARSA expression was markedly higher compared to the gray matter and was predominantly localized in the processes and bodies of glial cells (Figure 9C–C2). Overall, ARSA expression in the lumbar spinal cord of the experimental group (with intrathecal AAV9-ARSA administration) was similar to that described above. However, single ARSA-overexpressing neurons were found in the ventral horns, which was not observed in the group with intravenous AAV9-ARSA administration or the control group (Figure 9D–D2).

An immunohistochemical analysis of the spinal ganglia at the levels of the cervical, thoracic, and lumbar spinal cord revealed the presence of ARSA-overexpressing neurons only in the experimental group with intravenous AAV9-ARSA administration (Figure 9E–E2). The number of ARSA-overexpressing neurons varied greatly within this group; however, no significant differences in this indicator were found between the spinal ganglia of different levels (the cervical, thoracic, and lumbar regions). An analysis of the spinal roots and the lateral femoral cutaneous nerve showed no difference in ARSA expression in the experimental and control animals (Figure 9F–F2). It is worth mentioning that the ARSA expression in the porcine PNS was originally significantly higher than that in the CNS. Accordingly, the possibility of AAV9-ARSA penetration into the PNS from the subarachnoid space could not be assessed in the observed conditions.

## 3. Discussion

MLD is a group of hereditary monogenic diseases characterized by lysosomal dysfunctions resulting from the accumulation of an uncleaved substrate. Currently, gene and gene–cell therapy are considered promising approaches for the treatment of MLD. Gene–cell therapy utilizes genetically modified cells transduced with retro- and lentiviruses [44,45]. A clinical study (NCT01560182) showed that the transplantation of CD34+ hematopoietic stem cells (HSCs) with ARSA overexpression into patients with pre-symptomatic or very early symptomatic MLD resulted in remyelination and the normalization of motor activity. However, one of the nine patients who already had symptoms of the disease at the time of transplantation did not experience an improvement in motor activity [33]. An extension of this study (NCT03392987) showed a favorable profile in patients; however, those with a rapid progression did not receive an effective treatment [46]. In 2021, Arsa-cel was approved in the EU under the name Libmeldy. Despite these promising results, there are still a number of challenges, including the long-term monitoring of efficacy and safety in patients, as retro- and lentivirus-based drugs are at risk of insertional mutagenesis [47]. Genetically modified HSCs are also known to have a therapeutic effect using cross-correction mechanisms [48].

Gene therapy shows great promise for treating LSDs. ARSA encoding different AAV serotypes could be a potential therapy for MLD, as AAVs can transduce neurons in a wide range from the injection site through anterograde neuronal transport [36]. AAVs have several attractive features as vectors, such as low immunogenicity compared to other vectors, broad tropism, and the ability to transduce neurons over a wide range from the injection site. An AAV can be easily constructed as a gene delivery vector by replacing the viral genome with the therapeutic cassette. Different AAV serotypes have different tropisms for different tissues. Among all the identified and characterized serotypes, AAV9 has the highest tropism for CNS and can cross the BBB [49,50].

AAV9’s efficacy has been demonstrated in various preclinical models of CNS disorders and in some clinical studies [51]. The intrathecal administration of AAV9 ensures transgene distribution throughout the nervous system. When administered intravenously, AAV9 can cross the BBB and enter the CNS [52]. The effectiveness of the intravenous administration of rAAV9 was shown in a mouse model [53], and the effectiveness of the intrathecal administration of AAV9-ARSA was shown in 6-week-old mice; however, no improvement in motor behavior was observed in 1-year-old mice [54]. In our study, the intravenous and intrathecal administration of recombinant AAV9 containing a unique codon-optimized sequence of the human *ARSA* gene were compared and analyzed for their efficacy and safety in minipigs.

We produced recombinant AAV9-ARSA and confirmed in vitro ARSA overexpression using an enzyme activity assay and a Western blot analysis. We then evaluated the ability of AAV9-ARSA to synthesize a functionally active enzyme after intrathecal or intravenous administration in minipigs. Following the intrathecal administration of AAV9-ARSA, we observed an increase in ARSA enzymatic activity in the CSF, which was consistent with other studies [55,56]. However, no statistically significant increase was found in CNS organs, which we suggest could be due to the large volumes of minipigs organs. Another possible reason is that different sites were selected for analysis during organ sampling and the acquired sites did not reflect the full picture of the enzymatic activity changes. The RT-PCR results showed overexpression in the cerebellum and the cervical, thoracic, and lumbar spinal cord following intrathecal administration. IHC also showed overexpression in the cerebellum and the ventral horns of the lumbar region as well as in neurons in the spinal cord, but no statistically significant differences were found between the groups. Other researchers have shown the transduction of CNS and PNS neurons following the intrathecal administration of AAV9-GFP to mice [57,58]. In our study, overexpression was not observed in the PNS.

We hypothesize that the differences in promoter usage, injection site, anatomical variations among experimental animals, and AAV vector design may have contributed to the observed discrepancies. Rachel M. Bailey et al. used a self-complementary AAV vector [57] and reported increased enzymatic activity in the cerebellum and the cervical and thoracic spinal cord following the intravenous administration of AAV9-ARSA. The RT-PCR analysis demonstrated ARSA overexpression in the thoracic spinal cord and the spinal ganglia of the cervical, thoracic, and lumbar regions. The IHC analysis also revealed overexpression in the cerebellum and the spinal ganglia. Our findings are supported by studies conducted by Yingqi Lin et al., where they demonstrated broad expression of the transgene in various organs of minipigs following the intravenous administration of AAV9-GFP, with overexpression primarily observed in various regions of the brain, without any adverse inflammatory reactions [59].

In order to evaluate the safety of our research, we conducted a biochemical blood test, cytokine profiling in blood serum, and a pathomorphological analysis. The high-dose intravenous administration of AAV9 has been reported to cause systemic and sensory neuron toxicity in non-human primates [60]. Rosenberg et al. also showed that the intraparenchymal administration of a higher dose of 1.5 × 10^12^, compared to 2.4 × 10^9^, was associated with local pathological changes at the injection site of the AAVrh.10 hARSA vector, along with some minor pathological changes in the spinal cord [61]. However, in our study, the biochemical analysis showed no increases in the levels of AST, ALT, total bilirubin, or creatinine-J following both the intrathecal and intravenous administration of AAV9-ARSA. The cytokine profile analysis of the minipigs’ CSF revealed increases in the IL-1ra levels following both the intrathecal and intravenous administration of AAV9-ARSA. IL-1ra is induced in response to various forms of stress, such as excess excitatory neurotransmitters during seizures, infections and inflammation, and neurotrauma [62]. It can be synthesized in the CNS by neurons, microglia, and infiltrating macrophages [63]. The increases in the IL-1ra levels observed in our study could have been due to the weekly lumbar punctures performed for CSF sampling, which may have caused inflammation and increased the IL-1ra levels to provide an anti-inflammatory response. The pathomorphological analysis did not reveal any abnormalities, and although it was not performed for the liver, the biochemical analysis showed no evidence of hepatotoxicity.

We have demonstrated that both the intravenous and intrathecal administration of AAV9-ARSA into minipigs resulted in increases in enzymatic activity in the CNS. The intrathecal administration led to the transduction of CNS cells and the expression of a functionally active ARSA enzyme. The intravenous administration, on the other hand, resulted in overexpression in the cerebellum, suggesting virus penetration into the CNS from the periphery. While increases in the enzymatic activity of ARSA in various parts of the CNS following the intravenous administration of AAV9-ARSA were demonstrated, the immunocytochemical analysis did not confirm these findings. In addition, the intravenous administration of AAV9-ARSA resulted in the transduction of spinal ganglia neurons in the PNS, which was not observed with the intrathecal administration of the same construct. This suggests that the intravenous delivery of AAV9 may be useful for the treatment of the PNS but requires higher doses of the vector compared to intrathecal administration, which is more invasive but uses lower doses.

## 4. Materials and Methods

### 4.1. Genetic Construct Design and Analysis

The OptimumGene algorithm was employed for the optimization of the ARSA gene codon composition, taking into consideration various factors that could potentially affect gene expression levels. These factors included the codon shift, the GC composition, the CpG dinucleotide content, the mRNA secondary structure, tandem repeats, restriction sites interfering with cloning, premature polyadenylation sites, and additional minor ribosome binding sites. The nucleotide sequence of the CDS mRNA of the human ARSA gene (GeneBank #NM_001085425.3, 1530 bp) was utilized as a matrix for codon optimization. Following codon optimization, the CDS mRNA of the ARSA gene was cloned into the pAAV-MCS plasmid vector (Agilent Technologies, Santa Clara, CA, USA) using recombinase at EcoRI/BamHI restriction sites. Codon optimization, the de novo synthesis of the CDS mRNA nucleotide sequence of the ARSA gene, and its cloning into the pAAV-MCS plasmid vector were performed by GenScript (NJ, USA). The correct assembly of the genetic construct was verified through a restriction analysis using BamHI restrictase (#ER0051, Thermo Fisher Scientific Inc., Logan, UT, USA) and sequencing.

### 4.2. Production of Preparative Amounts of Plasmid Constructs Required for AAV Assembly

The TOP10 strain of Escherichia coli (Invitrogen, Waltham, MA, USA) was utilized to generate preparative amounts of plasmids through transformation. Cells were incubated in an LS-LB medium without antibiotics. Competent cells were prepared using the CaCl_2_ method. The genetic transformation of competent cells was performed using a heat shock, after which the transformed bacterial cells were incubated on a selective medium containing ampicillin. Plasmid DNA (pAAV-ARSA, pAAV-RC, and pHelper) was isolated from the resulting bacterial biomass (GeneJET Plasmid Miniprep Kit, #K0502, Thermo Fisher Scientific Inc., Logan, UT, USA).

### 4.3. Preparation and Purification of Recombinant AAV

AAV viruses were produced using the standard co-transfection of three plasmids into AAV293 cells via the calcium phosphate method. AAV293 cells were cultured at 37 °C in humid conditions with 5% CO_2_ in a complete DMEM medium (PanEco, Moscow, Russia) containing 10% fetal calf serum, L-glutamine, and 1% antibiotics such as penicillin and streptomycin. Cells were centrifuged 72 h post-transfection, after which a lysis buffer (NaCl, Tris-HCl (pH 8.5), MgCl_2_, and dH_2_O), DNase (Benzonase^®^ Nuclease, Sigma-Aldrich, St. Louis, MI, USA), and 25% sodium deoxycholate were added to the pellet. The lysates were purified and subjected to an iodixanol density gradient of 60%, 40%, 25%, and 15%. In the final stage of virus purification, a concentrator suitable for 50 kDa proteins (Vivaspin 20, 50 kDa membrane, Sartorius, Havant, UK) was utilized. The viral titer was determined through quantitative PCR using primers (forward: 5′5-3′-GGAACCCCTAGTGATGGAGTT and reverse: 5′-3′-CGGCCTCAGTGAGCGA) and a probe targeting ITRs (5′-3′ (FAM) CACTCCCTCTCTGCGCGCTCG (BBQ).

### 4.4. Determination of the Overall Purity of Virus Particles

The overall purity of viral particles was determined using sodium dodecyl sulfate–polyacrylamide gel protein electrophoresis (SDS-PAGE). A polyacrylamide gel was prepared with an acrylamide concentration gradient (4% for the concentrating gel and 10% for the separating gel). AAV9-ARSA was incubated at 70 °C for 15 min prior to loading onto the gel. A protein marker (Thermo Scientific™, Cat. No. 26616) and the sample were added to their corresponding wells, and electrophoresis was performed at a constant voltage of 200 V.

After electrophoresis, the gel was stained with Coomassie blue (Thermo Fisher, Cat. No. LC6065) for 2–3 h with gentle shaking. The gel was then washed in a solution of 40% methanol and 10% acetic acid until protein bands were visible. Finally, an image was captured using the ChemiDocXRS+ gel documentation system (BioRad, Hercules, CA, USA).

### 4.5. Western Blot Analysis

For the transfection of HEK293T cells with recombinant AAV9-ARSA, cells were seeded in a 6-well plate 24 h prior. A mixture was prepared, considering a multiplicity of infection (MOI) of 100 (100 viral particles per cell), with protamine sulfate added to achieve a final concentration of 10 μg/mL. The cell growth medium was replaced with the prepared transduction mixture. After 6 h, the medium was switched with a fresh medium. The transgene expression was assessed using a Western blot analysis.

The Western blot analysis was performed using the standard Laemmli method under denaturing conditions (SDS-PAGE). The protein transfer from gel to membrane (PVDF) was conducted using a Trans-Blot^®^ SD Semi-Dry Electrophoretic Transfer Cell (BioRad, Hercules, CA, USA). Non-specific binding was prevented by incubating the membrane in a blocking buffer. The membrane was then incubated with primary rabbit anti-ARSA monoclonal antibodies (#PAG619Hu01, Cloud-Clone Corp., Houston, TX, USA), followed by incubation with secondary antibodies (polyclonal goat antibodies to human immunoglobulin G conjugated with horseradish peroxidase, Sigma, #A6154, USA). The visualization of the immunoprecipitate was carried out using ECL Western Blotting Substrate (#W1001, Promega, Madison, WI, USA). The membrane was examined using the ChemiDocXRS+ gel documentation system (BioRad, Hercules, CA, USA).

### 4.6. Animals

All experimental procedures on animals were reviewed and approved by the local ethics committee of Kazan Federal University (protocol No. 23, 30 June 2020). Healthy female minipigs at the age of 4 months (weighing 9–12 kg) were used in this study. The experimental animals (15 minipigs) were divided into 3 groups: (1) intrathecal administration of AAV9-ARSA at a dose of 1 × 10^12^ genomic copies (gc)/kg (n = 5); (2) intravenous administration of AAV9-ARSA at a dose of 3.77 × 10^13^ gc/kg (n = 5); and (3) a control group with no AAV9-ARSA administration (n = 5). Animals were housed in specialized animal care facilities of Kazan State Academy of Veterinary Medicine, named after N.E. Bauman (KGAVM), under the supervision of qualified personnel. Animal euthanasia was performed in strict compliance with the recommendations for the euthanasia of experimental animals of the European Commission.

### 4.7. Material Sampling

Prior to the virus injection, CSF and blood samples were taken at time point zero. Later, CSF and blood samples were taken 7, 14, 21, 28, and 35 days following the virus injection in order to analyze the dynamics of the ARSA enzymatic activity. On the 35th day after the virus injection, the experimental animals were euthanized. To determine the ARSA expression levels, quantitative RT-PCR was performed, as was a test for ARSA enzymatic activity in the homogenates of various parts of the nervous system.

Parts of the following organs were sampled from each animal: the cerebellum; the occipital lobe of the brain; cervical (C6-7), thoracic (Th6-7), and lumbar (L2-3) sections of the spinal cord with spinal roots and ganglia; the lateral femoral cutaneous nerve, the heart (left ventricle), the lungs, the kidneys, the spleen, and the liver. Each of the organ parts was placed in a 10% formalin solution. After 48 h of fixation, each organ part was divided into 2 fragments; one was embedded in paraffin, and the other was transferred consecutively into 15% and then 30% sucrose. Samples obtained from minipigs’ nervous tissues (in 30% sucrose) were placed in a tissue freezing medium (Tissue-Tek O.C.T. Compound, Sakura, Torrance, CA, USA). On a Microm HM 560 cryostat (Thermo Scientific), 20 μm thick transverse or sagittal sections of the studied nervous system organs were obtained and used for a subsequent immunofluorescence analysis. Samples of internal organs embedded in paraffin were cut into 5–7 µm thick sections on a Minus S700A (RWD) microtome and then stained with hematoxylin and eosin.

### 4.8. Determination of ARSA Enzymatic Activity

To determine the ARSA enzymatic activity in organ homogenates, the total protein concentrations in samples were determined using a Pierce™ BCA Protein Assay Kit (ThermoFisher Scientific Inc., USA). The samples were normalized according to the total protein concentration. Next, 50 µL of cell lysate or a tissue homogenate sample was incubated with a solution of a nitrocatechol sulfate substrate (0.01 M p-Nitrocatechol sulfate dipotassium salt (#N7251, Sigma-Aldrich), 0.5 M sodium acetate, 5 × 10^−4^ Na_4_P_2_O_7_, 10% sodium chloride, pH = 5) for 1 h at 37 °C. The reaction was then stopped by adding 1 N sodium hydroxide. Sulfatase dilutions (#S9626, Sigma-Aldrich) were used as standards, and the optical density was measured at a wavelength of 515 nm.

### 4.9. Quantitative Polymerase Chain Reaction (qPCR)

Total RNA was isolated from animal organ parts using the TRIzol Reagent (Invitrogen, Waltham, MA, USA) following the manufacturer’s protocol. Primers and probes specific to human ARSA were designed using the GenScript Online Real-time PCR (TaqMan) Primer Design Tool (GenScript, Piscataway, NJ, USA) and synthesized by Evrogen (Moscow, Russia). The primer sequences were as follows: forward: 5′-CAAGGTACATGGCATTCGCA-3′ and reverse: 5′-CTGTGGATAGTGGGTGTGGT-3′. The probe sequence was 5′-CCTGCCGCTGTGCATCTGCCA-3′ labeled with FAM (6-carboxyfluorescein) on the 5′ end and BHQ-1 (Black Hole Quencher 1) on the 3′ end. Isolated RNA was reverse-transcribed into cDNA using the GoScript™ Reverse Transcription System (Promega, Madison, WI, USA) according to the manufacturer’s instructions. Real-time PCR based on TaqMan was performed in 96-well MicroAmp plates (BioRad, Hercules, CA, USA) using a qPCR mix that contained 1 µL of a cDNA template, 0.3 µL of a primer and probe mixture (with a final concentration of 300 nM for each primer), 4.7 µL of MilliQ H_2_O, and 4 µL of 10× TaqMan buffer (Lytech, Moscow, Russia), with a final volume of 10 µL. PCR amplification was carried out using the CFX96 Touch™ Real-Time PCR Detection System (BioRad, Hercules, CA, USA) under the following temperature cycling conditions: preheating at 95 °C for 3 min, 45 denaturation cycles at 95 °C for 10 s, and annealing at 55 °C for 30 s.

### 4.10. Biochemical Blood Analysis and Cytokine Profile Analysis

To evaluate safety, a biochemical blood analysis and cytokine profiling were performed. Whole blood was collected from animals into test tubes with gel and a blood clotting activator. The samples were then centrifuged at 1900 rpm for 20 min to separate the serum, and the levels of aspartate aminotransferase (AST), alanine aminotransferase (ALT), total bilirubin, and creatinine were measured using a ChemWell 2900 biochemical analyzer (USA).

Porcine serum and CSF samples were analyzed using the MILLIPLEX MAP Porcine Cytokine/Chemokine Magnetic Bead Panel-Immunology Multiplex Assay (PCYTMG-23K-13PX, Merck), which included GM-CSF (colony-stimulating factor 2 (granulocyte-macrophage)), IFNγ (interferon gamma), IL-1α (interleukin 1 alpha), IL-1β, IL-1ra (IL-1 antagonist), IL-2, IL-4, IL-6, IL-8, IL-10, IL-12, IL-18, and TNF-α (tumor necrosis factor-alpha). For each sample, 50 microliters were used to determine the analyte concentrations. The data were analyzed using a Luminex 200 analyzer with MasterPlex CT control software v.3 and MasterPlex QT analysis software v.3 (MiraiBio division of Hitachi Software, Irvine, CA, USA).

### 4.11. Immunofluorescence Analysis

To analyze the tissue expression of the ARSA protein, transverse or sagittal cryostat sections of CNS and PNS organs obtained from minipigs were used. For immunofluorescent labeling, sections were blocked with 5% normal goat serum then incubated with a primary antibody (anti-ARSA, Cloud-Clone, Tokyo, Japan) and subsequently incubated with secondary antibodies (Alexa 546, Abcam, Cambridge, UK). Following successive washes in PBS, sections were counterstained with 4′,6-diamidino-2-phenylindole (DAPI) (10 μg/mL in PBS, Sigma-Aldrich) to visualize nuclei. Sections were mounted with a medium (ImmunoHistoMount, Santa Cruz, CA, USA) and examined using an LSM 700 confocal scanning microscope (Carl Zeiss, Jena, Germany).

### 4.12. Statistical Analysis

The obtained data were analyzed using GraphPad Prism 9.3.1 software (GraphPad Software). The Shapiro–Wilk test and a one-way analysis of variance (ANOVA) followed by Tukey’s post hoc test were applied to determine statistically significant differences, which were designated as * for *p* < 0.05, ** for *p* < 0.01, *** for *p* < 0.001, and **** for *p* < 0.0001.

## Figures and Tables

**Figure 1 ijms-24-09204-f001:**
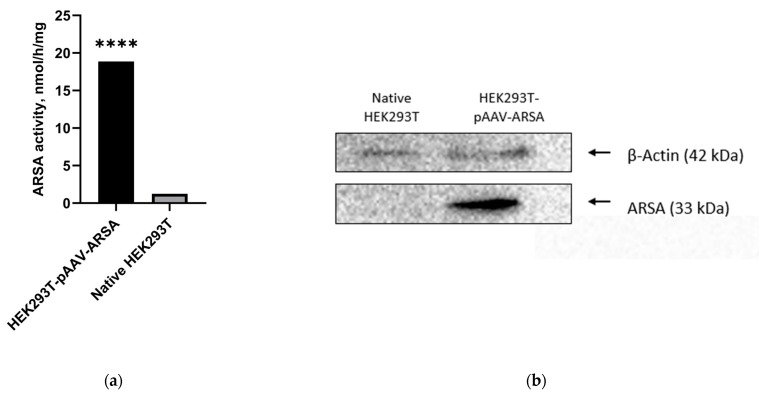
(**a**) Analysis of ARSA enzymatic activity in HEK293T cell lysate after pAAV-ARSA transfection. (**b**) Western blot analysis of ARSA proteins in native and genetically modified HEK293T pAAV-ARSA cells. Data are presented as means ± S.D. ****—*p* < 0.0001.

**Figure 2 ijms-24-09204-f002:**
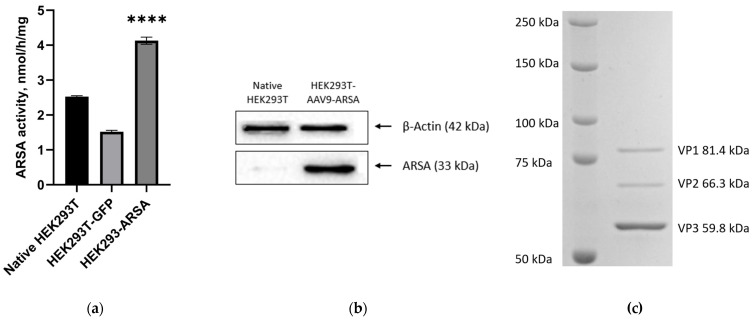
(**a**) Analysis of ARSA enzymatic activity in HEK293T cell lysates following AAV9-ARSA transduction. (**b**) Western blot analysis of ARSA proteins in native and genetically modified HEK293T AAV9-ARSA cells. (**c**) Determination of overall purity of AAV-ARSA using SDS-PAGE protein electrophoresis, where VP1, VP2, and VP3 are viral proteins. Data are presented as means ± S.D. ****—*p* < 0.0001.

**Figure 3 ijms-24-09204-f003:**
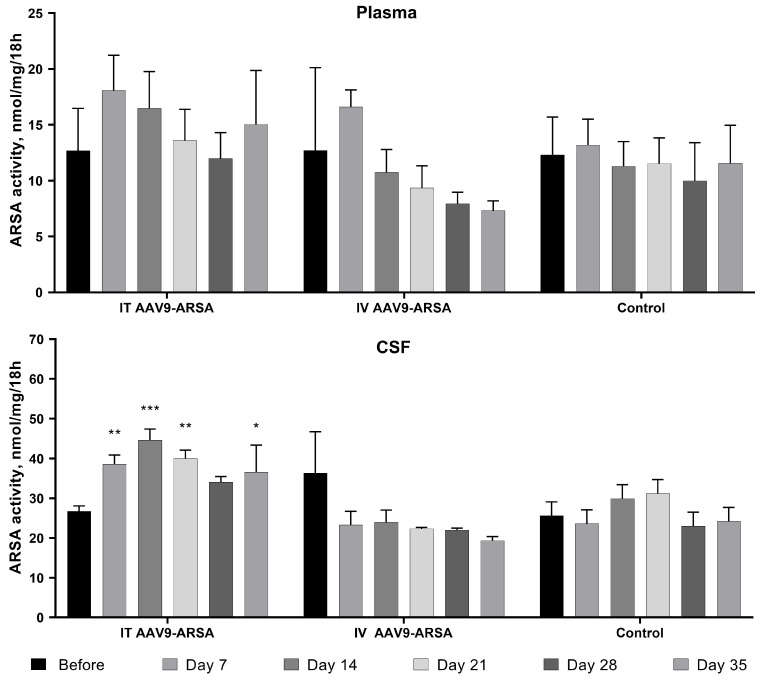
Analysis of ARSA enzymatic activity in plasma and CSF of minipigs following AAV9-ARSA administration. Data are presented as means ± S.D. *—*p* < 0.05, **—*p* < 0.01, ***—*p* < 0.001. IT AAV9-ARSA—intrathecal administration of AAV9-ARSA; IV AAV9-ARSA—intravenous administration of AAV9-ARSA.

**Figure 4 ijms-24-09204-f004:**
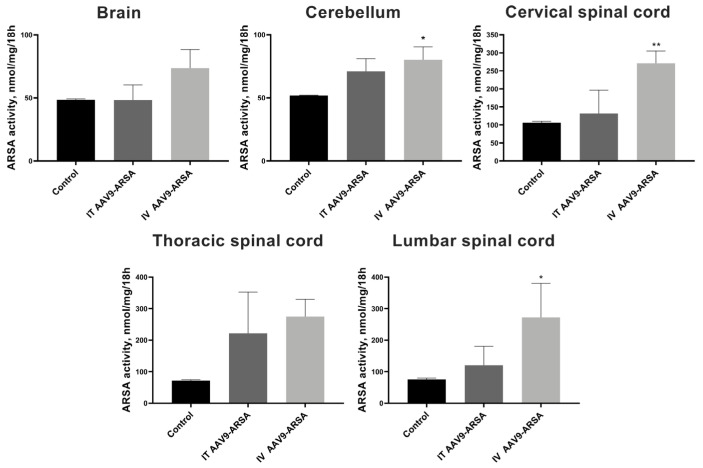
Analysis of ARSA enzymatic activity in homogenates of various parts of the minipigs’ nervous systems following the administration of AAV9-ARSA. Data are presented as means ± S.D. *—*p* < 0.05, **—*p* < 0.01. Control—control group of animals; IT AAV9-ARSA—intrathecal administration of AAV9-ARSA; IV AAV9-ARSA—intravenous administration of AAV9-ARSA.

**Figure 5 ijms-24-09204-f005:**
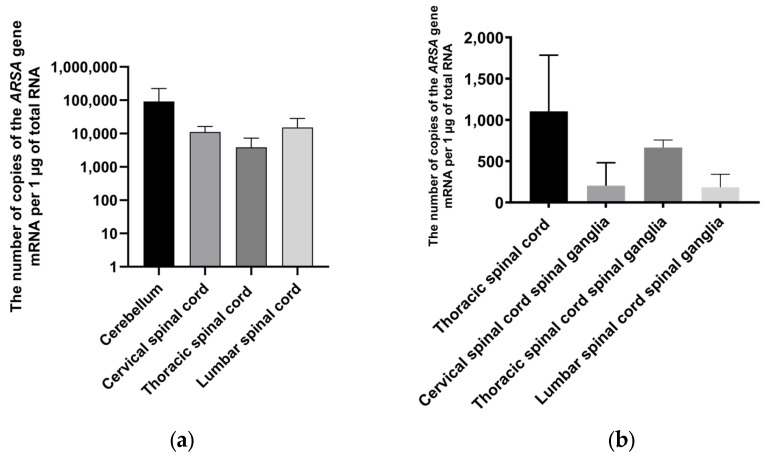
Copy numbers of *ARSA* gene mRNA in different parts of the nervous systems of minipigs on the 35th day following AAV9-ARSA administration. Data obtained via quantitative PCR. Data are presented as means ± S.D. (**a**) following the intrathecal administration of AAV9-ARSA and (**b**) following the intravenous administration of AAV9-ARSA.

**Figure 6 ijms-24-09204-f006:**
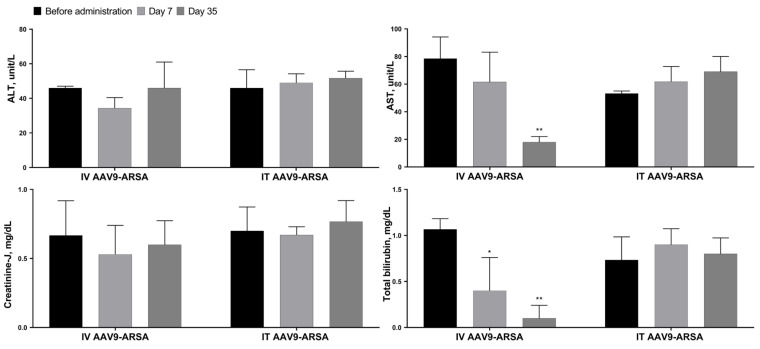
Biochemical parameters of the minipigs’ blood sera following AAV9-ARSA administration. IT AAV9-ARSA—intrathecal administration of AAV9-ARSA, IV AAV9-ARSA—intravenous administration of the AAV9-ARSA vector. * *p* <0.05, ** *p* < 0.01.

**Figure 7 ijms-24-09204-f007:**
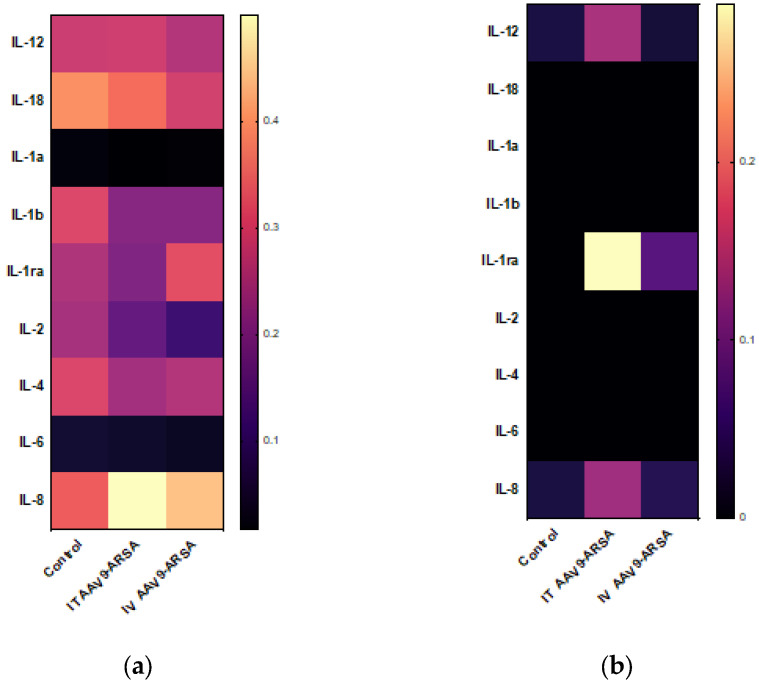
Analysis of inflammatory cytokine and chemokine levels following intrathecal and intravenous administration of AAV9-ARSA. (**a**) Cytokine profile of porcine blood serum. (**b**) Cytokine profile of porcine CSF. IT AAV9-ARSA—intrathecal administration of AAV9-ARSA, IV AAV9-ARSA—intravenous administration of AAV9-ARSA.

**Figure 8 ijms-24-09204-f008:**
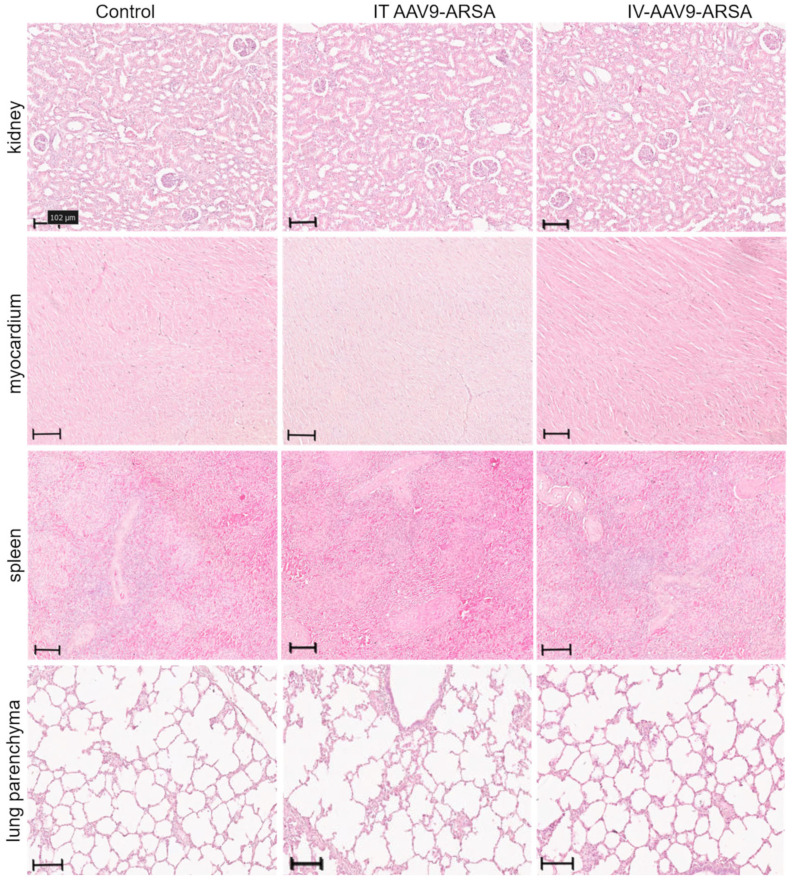
Pathological analysis of kidney, heart, spleen, and lung tissues in the control and experimental groups on the 35th day following AAV9-ARSA administration. Staining with hematoxylin and eosin. Control—control group of animals, IT AAV9-ARSA—intrathecal administration of AAV9-ARSA, IV AAV9-ARSA—intravenous administration of the AAV9-ARSA vector.

**Figure 9 ijms-24-09204-f009:**
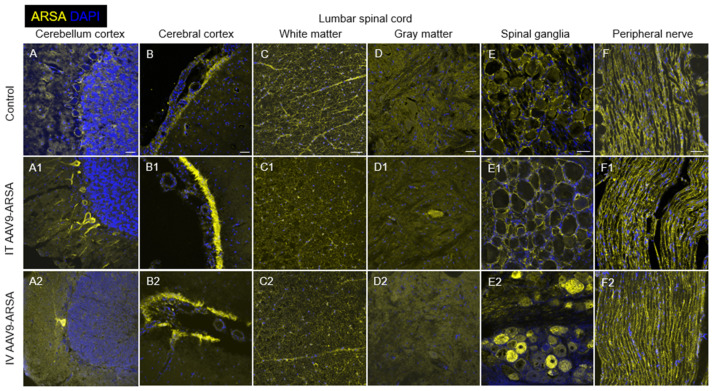
Assessment of ARSA (yellow) expression in different regions of nervous tissue in the control group of animals (**A**–**F**) and in experimental groups 1 and 2 on day 35 after the intrathecal (**A1**–**F1**) and intravenous (**A2**–**F2**) administration of AAV9-ARSA, respectively. Nuclei were stained with DAPI (blue). Scale bar: 50 µm. Control—control group of animals, IT AAV9-ARSA—intrathecal administration of AAV9-ARSA, IV AAV9-ARSA—intravenous administration of AAV9-ARSA vector.

## Data Availability

Data sharing applicable.

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
