# Peer review of "Safety and Efficacy of Intravenous and Intrathecal Delivery of AAV9-Mediated ARSA in Minipigs"

_ijms, 2023, doi:10.3390/ijms24119204_

Round 1
Reviewer 1 Report
The manuscript compares the use of an AAV vector encoding for the ARSA gene, deficient in ML, using either IT or IV injection, in minipigs. The authors provide biochemical, molecular and histological data on vector biodistribution and safety.
Some concerns:
- The dose of vector using IT and IV is different. I understand why it was done this way, but did the authors perfprm another group using the same dose? It would be usefu for direct comparison.
- It is interesting that IT injection lead to higher CSF ARSA activity and really high copy numbers, but lower brain tissue activity, compared to intravenous group. Any ideas on why does that happen? Again, a same-dose group could help in comparing the results.
-
Author Response
Thank you for your review. Unfortunately, we did not conduct a study using the same dose, but different routes of administration. An appropriate dose of the vector was selected after literature analysis, focusing on the biodistribution of the vector and treatment’s safety. The choice of a lower dose for intrathecal administration was decided to avoid developing an acute immunological reaction in response to high doses of intrathecal AAV, as described in the literature. Side effects of high doses of intrathecal AAV were described in the clinical study NCT03381729.
We assume the reason could be the large volume of minipigs organs. Moreover, different sites were selected for analysis during organ sampling, and the acquired site does not reflect the full picture of enzymatic activity changes.
Author Response
Thanks for your comments, all comments have been revised. The title of the manuscript will be changed to "Safety and Efficacy of Intravenous and Intrathecal Delivery of AAV9-mediated ARSA in Minipigs"
Reviewer 3 Report
The present paper from Mullagulova and colleagues reports the results of two AAV9-based gene transfer procedures to deliver the arylsulfatase A (ARSA) cDNA to minipigs. The final aim was to assess safety and efficacy of intravenous (IV) and intrathecal (IT) administration of the viral vector and of gene expression, in a large animal model such as minipig, in view of the development of human therapy for Metachromatic leukodystrophy (MLD), a rare inherited lysosomal storage disease due to mutations in the ARSA gene.
In recent years, several published works have attempted to develop gene therapy treatment for this severe genetic condition, affecting the CNS and peripheral nervous system. This manuscript presents an additional afford to improve the treatment, designing a codon optimized ARSA cDNA, which is significantly more expressed than the wild type version, and exploring the more efficient AAV9 gene transfer route to relevant nervous tissues in a large animal that in several anatomical structures closely mimic human anatomy and, specifically, neural complexity.
Overall, the findings presented in this manuscript are similar to those previously described by others.
The results obtained from the characterization of the transduction efficiency of AAV9-ARSA through the two route of administration show, however, some incongruities that need clarification.
Despite quantitative PCR analysis, using specific primers to detect human ARSA gene expression, revealed more abundant mRNA amount in nervous structure such as cerebellum, cervical, thoracic and lumbar spinal cord, after IT compared to IV administration route, the ARSA enzymatic activity displayed significant higher values above baseline only in the group of animals injected IV.
The immunofluorescence analysis also seems to indicate that intrathecal administration favors a more widespread and efficient transduction in the brain.
Unfortunately, since there are no pig models of MLD disease, efficacy of the gene therapy treatment cannot be proved in the present settings.
Nonetheless, based on clinical data and previous findings on gene therapy research, it would be interesting to anticipate which of the two routes of administration, or both, has more potential to produce a therapeutic effect in patients.
Recent GT clinical trial in MLD patients with AAVrh10 was unsuccessful, and therefore it is important to establish what transduction efficiency levels and cellular targets must be most relevant for an AAV-based treatment to prevent or halt disease progression.
Other criticisms:
Figure 1 - The description of the cloning is not appropriate for results section and can be moved to supplementary or methods.
Figure 2b - The band of endogenous ARSA protein expressed by Native HEK293Tcells is not visible, choose a more representative image.
The authors should update reference citations and broaden the discussion
For example, a paper of Miyake et al., published in 2021, reports a complete rescue of the pathological phenotype of MLD mice intratechally injected with an AAV9-ARSA vector.
Additionally, the group of Aiuti published an important report on long term follow-up of safety and efficacy of Arsa-cell clinical trial (Fumagalli et al. 2022), a procedure that already obtained full marketing authorization in Europe, and that need to be included in the discussion.
Author Response
Thanks for your comments, all comments have been revised.
We think that intrathecal administration has a greater potential for the treatment of MLD, as myelin degeneration occurs in the central nervous system. Intrathecal administration allows the viral vector to be delivered directly into the fluid surrounding the brain and spinal cord.
New studies were added to this review. ARSA-cell was mentioned in discussion, 1 paragraph, and clinical trial NCT01560182, but we added Fumagalli et al. 2022 to the bibliography.
Round 2
Reviewer 3 Report
The authors addressed my suggestions and made some changes that improve the present version of the manuscript.